# C2Rust-Bench: A Minimized, Representative Benchmark for C-to-Rust Transpilation

## Abstract

Despite significant effort in vulnerability detection over the last two decades, memory safety vulnerabilities continue to be a systemic problem that affects most mainstream software. Recent reports have concluded that the key to solving this issue once and for all is to migrate legacy C code to memory-safe languages. To this end, C-to-Rust "transpilation" has become a popular research topic. Recent work has proposed various approaches; however, what the community lacks is a comprehensive evaluation dataset. Currently, researchers rely on completeness through sheer sample volume, but this bloats the time required to run experiments and makes verification, which is currently done manually, laborious. In this work, we propose a method for selecting functions from a large set to construct a minimized yet representative dataset to evaluate C-to-Rust transpilation systems. We propose C2Rust-Bench, a dataset of only 2,905 functions that are nevertheless an objectively representative benchmark for C-to-Rust transpilation. This dataset was distilled from 15,503 real-world functions encompassing previous work.

## 1 Introduction

The recent advancements in Artificial Intelligence (AI) have spurred discussions of code migration (aka. "transpilation") from one programming language to another. Certainly, transpilation of C programs into Rust is one of the most popular because Rust provides memory-safety by design while maintaining comparable runtime performance. It is reported that 70% of the Common Vulnerabilities and Exposures (CVEs) assigned by Microsoft relate to memory safety (CISA, 2025). Instead of relying on detection and mitigation, the US White House Office of National Cyber Directory (ONCD) have concluded that migration to memory-safe languages is the most promising solution (WhiteHouse, 2025), leading to the announcement of funding for C-to-Rust transpilation research (DARPA, 2025).

Initial attempts have been made to propose automated tools to translate C programs into Rust using Large Language Models (LLMs) (Emre et al., 2021; Zhang et al., 2023; Yang et al., 2024; Hong & Ryu, 2024; Shiraishi & Shinagawa, 2024). However, there is a non-trivial question: *What dataset should be used to evaluate the proposed tools?* As observed in existing program analysis work, such as vulnerability detection, an evaluation dataset should be, most importantly, concise and representative. By being concise, a dataset reduces the time spent evaluating a proposed system, and by being representative, the dataset reinforces the validity of the findings. Failing to satisfy these properties results in costly and laborious evaluations that hinder research progress.

To our knowledge, no prior work provides an objectively concise and representative benchmark for evaluating C-to-Rust transpilation. Concurrently, another work (Khatry et al., 2025) proposes a repository-scale benchmark, derived from previous studies, with manually crafted Rust interfaces and test suites for end-to-end evaluation; however, it does not offer a representative, minimized set for initial assessment. While other security topics, such as vulnerability detection, have such benchmarks (Dolan-Gavitt et al., 2016; Hazimeh et al., 2020), they do not aim for conciseness for domain-specific reasons. In contrast, approaches of existing work in machine learning (ML) (Bachem et al., 2017; Sener & Savarese, 2017; Novikov et al., 2021; Lee et al., 2024; Song et al., 2025), reducing training dataset size, are tailored to generic text and images, which makes them poorly suited for use on software source code.

Reducing a large dataset of real-world C programs without losing distinctive features poses several challenges. Programming constructs in a language can be thought of as small in number; however,

the combinations of those constructs produce unique programs with varying complexity levels. This poses the first challenge, which is that we have a nearly infinite number of programs to select from. Moreover, even with a finite set of programs, it would still be difficult to select representative programs because how they relate to the task of transpilation is unclear. We require an answer to the questions: *What is a representative C program set for transpilation and how can we describe it?*

Even though combinations of programming constructs can create an infinite number of programs, we observe that LLM and rule-based transpilation systems break down larger programs into discrete finite units for processing. In short, there is an upper bound beyond which increasing the size of a program no longer increases the complexity of transpilation, and, by extension, representativeness. Moreover, representativeness can be defined as having a subset of programs that contains all the challenging code patterns contained in the larger original set. Based on this definition and our observations, we aim to obtain a set of metrics that express the complexity of a given program, which overcomes the challenge of lack of quantitative metrics for source code.

Based on our insights, we propose a method that takes a large dataset from real-world programs from various domains and down-selects based on complexity metrics. In the literature, there are various software metrics that represent different features of a given code. We find that the Maintainability Index (MI) suits our purpose well (Coleman et al., 1994) since it is the summary of three other metrics, including (1) cyclomatic complexity (McCabe, 1976), representing control flow complexity, (2) Halstead's volume (Halstead, 1977), representing the amount of information contained within a code, and (3) source lines of code, representing the size of the code. Since our target task is C-to-Rust transpilation and the two languages differ in several aspects, such as memory operations and data types, taking only C code into account for measuring complexity would be limiting for capturing representativeness. Thus, in addition to MI of C code, we cover the MI of Rust code as a separate complexity metric. Moreover, we observed that it is challenging to transpile C code containing pointer arithmetic, memory operations, and advanced data structures. Based on those observations, we identify two other sources that represent the complexity of Rust code for the transpilation task, namely, the usage of unsafe code and the usage of varying data types.

In the selection process, we utilize partitioning by cutting each metric into pieces whose combinations form bins in multidimensional space. Then, we calculate the principal component analysis (PCA) complexity score, which is the summary of four metrics, and order each bin by the PCA complexity score. Lastly, we perform selection using systematic sampling from each bin to obtain representative samples from distinctive data points.

To form an initial large dataset to select from, we obtain the programs used in previous evaluations in C-to-Rust transpilation (Emre et al., 2021; Zhang et al., 2023; Yang et al., 2024; Hong & Ryu, 2024), which yields 65 programs containing 15,503 functions. Applying our selection process, we obtain 2,905 functions for our benchmark, C2RUST-BENCH, which is an 81.3% reduction in samples. We release C2RUST-BENCH to be used in the evaluation of the C-to-Rust transpilation works.[1] In addition, we publish the code artifacts of our implementation on GitHub.[2]

## 2 OVERVIEW

In this section, we first present dataset usage and potential challenges related to dataset in program analysis and machine learning fields. Then, we provide a brief background on the transpilation from C to Rust. Next, we explain our motivation behind the dataset reduction for transpilation.

### 2.1 BACKGROUND

**Dataset in program analysis.** Program analysis research typically involves developing tools for tasks such as vulnerability detection, malware detection, or transpilation. These tools must be evaluated on a set of programs datasets such as those containing known CVEs, malware samples, or code with diverse characteristics. Two primary approaches are commonly used to construct such datasets: (1) selecting real-world programs from varying domains (Emre et al., 2021) or (2) retrieving the dataset from previous work if it is available (Zhang et al., 2023). Some studies also combine collected set

---

[1]https://huggingface.co/datasets/anonymous4review/C2Rust-Bench
[2]https://github.com/anonymous8428/C2Rust-Bench

with prior datasets (Hong & Ryu, 2024). However, these approaches face challenges: (1) small datasets may fail to comprehensively evaluate the tool, (2) large datasets increase analysis time, or (3) using different datasets hinders tool comparison. This presents an optimization problem of identifying a minimized representative set with distinct samples.

**Dataset in Machine Learning.** ML models are built from data through training and testing processes. Thus, dataset collection and selection play a critical role, dealing with the trade-off between accuracy and training/testing time. Increasing the amount of data may improve accuracy but often adds duplicates, raising cost without benefit. As a result, several works in the ML literature focus on reducing dataset size by selecting a subset of distinct or representative data points from a large dataset (Bachem et al., 2017; Sener & Savarese, 2017; Novikov et al., 2021; Lee et al., 2024; Song et al., 2025). This reduction helps to reduce training and testing time while maintaining accuracy.

**Transpilation.** C-to-Rust transpilation has gained traction as a long-term solution to memory-safety vulnerabilities (WhiteHouse, 2025; DARPA, 2025). Initial works propose rule-based C-to-Rust transpilation tools such as c2rust (Immunant, 2025a). c2rust produces compilable and semantically correct Rust code; however, the output is not idiomatic and is wrapped in unsafe blocks by default. Subsequent efforts aim to improve c2rust's output by reducing unsafe usage (Emre et al., 2021; Zhang et al., 2023) and adjusting function returns with proper Rust types (Hong & Ryu, 2024). However, due to the inherent limitations of rule-based approaches, recent efforts explore AI-driven approaches, including the LLM-based transpiler VERT (Yang et al., 2024) and methods addressing the limits of the LLM context-window (Shiraishi & Shinagawa, 2024).

## 2.2 MOTIVATION

Despite advances in vulnerability detection (Song et al., 2019), memory safety vulnerabilities remain a critical threat (CISA, 2025; WhiteHouse, 2025). Rather than securing memory-unsafe programs, migrating them to memory-safe languages, called transpilation, is considered the best alternative solution (WhiteHouse, 2025; DARPA, 2025). While C language lacks memory safety by design, Rust provides a safer alternative, allowing unsafe operations only via the `unsafe` keyword. This makes the C-to-Rust transpilation crucial to mitigate memory risks (WhiteHouse, 2025; DARPA, 2025).

Several works have been done on the C-to-Rust transpilation, focusing on rule-based approach (Emre et al., 2021; Zhang et al., 2023; Hong & Ryu, 2024) and LLM-based approach (Yang et al., 2024; Shiraishi & Shinagawa, 2024), marking the inception of a new research area. However, as seen in other program analysis fields, determining an appropriate dataset to evaluate the proposed tools is challenging due to: (1) having a small dataset that leads to not evaluating the tool comprehensively, (2) having a large dataset that leads to a long analysis time, automated and manual analysis, or (3) having a different dataset from previous work that makes them incomparable.

To address these challenges in this emerging field, we form a benchmark set C2RUST-BENCH, containing 2,905 distinct and representative functions selected from 15,503 functions in the large set, reducing it by 81.3%. C2RUST-BENCH avoids the three potential problems previously mentioned by providing (1) a small but representative dataset that allows comprehensive evaluation, (2) a minimized dataset that reduces the analysis time by approximately 80%, and (3) a standardized dataset that offers a common ground for the comparison of incremental works.

## 3 METHODOLOGY

In this section, we first present an overview of our methodology. Then, we explain the source code complexity metrics. Lastly, we present the function selection method.

### 3.1 OVERVIEW

Selecting representative functions from a large dataset for evaluating C-to-Rust transpilation requires (1) identifying features that represent the functions with respect to the target task and (2) a method to select functions based on the features. We identify and define a set of source code complexity metrics for C and Rust. To obtain the Rust counterpart of a C function, we build a transpilation framework using a local LLM, enabling feature extraction from both sides of the C-to-Rust transpilation. We discuss the implications of using an LLM to construct an evaluation dataset for transpilation in §B.2.

Selecting functions based on code complexity metrics requires a method to identify distinctive data points in a multidimensional space. We use a partitioning method that divides each dimension into segments based on a specified partitioning parameter. Combinations of these segments in the multidimensional space form bins. Within each bin, we rank samples using Principal Component Analysis (PCA) scores derived from their code metrics and apply systematic sampling to select representative and diverse functions. The following subsections present this methodology in detail.

## 3.2 Representative Source Code Complexity Metrics

In this subsection, we describe the source code complexity metrics we collect or define to represent function complexity for the selection process.

**Maintainability Index.** One popular metric to measure the complexity of a source code is the Maintainability Index (MI) (Oman & Hagemeister, 1992). The MI metric is built on top of three other metrics, cyclomatic complexity (McCabe, 1976), Halstead's volume (Halstead, 1977), and source line of code. MI metric combines these three metrics to assess multiple dimensions of code maintainability, particularly how easily developers can understand, modify, and extend a code. Because transpilation requires carefully analyzing the original code and rewriting it in another language while preserving its behavior, MI provides a meaningful measure of the code's complexity in this context. Therefore, the MI metric serves as an excellent candidate for use in our selection process.

**Unsafe Code Complexity.** Memory operations such as pointer arithmetic and dereferencing pose significant challenges in C-to-Rust transpilation due to fundamental differences in how memory safety is handled in these languages. In C, these operations are unrestricted and rely entirely on the programmer for correctness. However, Rust enforces strict memory safety guarantees, allowing operations such as pointer manipulation only in explicitly marked unsafe contexts. As a result, transpilation must take into account these restrictions to generate a correct and safe Rust code.

Considering the previous points and our observations using LLM for transpilation, the use of unsafe code in Rust indicates a higher complexity of a code for transpilation. Since the MI index does not take memory operations into account in measuring maintainability, we define a metric to represent the complexity of a Rust code with respect to its unsafe operations. We obtain unsafe blocks and count the number of unsafe statements. Then, we find the average unsafe statement for a given Rust function. The details of the implementation of this metric are presented in §A.2.

**Data Type Complexity.** Data types pose a significant challenge in the C-to-Rust transpilation due to fundamental differences in type systems, memory layouts, and handling of basic and complex data types. In C, the type system is relatively permissive, allowing implicit type conversions, type casts, and low-level memory manipulations. For instance, C allows casting between incompatible types and perform arithmetic on raw pointers, which can result in memory misalignments or violations of type safety. In contrast, Rust enforces a stricter type system that prioritizes safety and explicitness. Rust disallows implicit type coercion and enforces explicit casting, requiring careful handling during transpilation. Additionally, concepts such as ownership, borrowing, and lifetimes—absent in C—complicate transpilation due to their impact on memory allocation and ownership.

Translating data types from C to Rust requires not only mapping flexible memory handling of C language to Rust's more restrictive model but also ensuring that the resulting code respects Rust's memory safety, ownership, and lifetime rules. This process requires accurately mapping both basic and complex types with appropriate conversions and safety checks. Given the complexity of this task, we introduce a metric that captures the type-related complexity of a Rust function based on the data types used. The details of the implementation of this metric are shared in §A.2.

## 3.3 Function Selection Method

We use 4 metrics to quantify the complexity of C and corresponding Rust functions: the MI metrics of C and Rust codes, and the unsafe code and data type complexity metrics of Rust code. To facilitate function selection, we need a method to collectively analyze these four metrics and identify representative data points capturing the characteristics of the entire set. While k-means clustering is a natural candidate for grouping multi-feature data, the continuous and gapless nature of our metric values makes such clustering approaches unsuitable.

For data with continuous values, a common practice is to divide the population into subgroups based on characteristics, as in stratified sampling. Since we do not have such categorical characteristics, we use an alternative but similar approach, partitioning, to create subgroups. We partition the value range of each metric into equal segments. In a multi-dimensional space, a combination of these segments across dimensions form bins, similar to the naturally formed subgroups in stratified sampling. Each of these bins represents a distinctive data point in the multidimensional space.

To obtain a representative set with distinct samples, we use systematic sampling combined with Principal Component Analysis (PCA) for selection from each bin. First, we calculate PCA for the samples in each bin, which condenses the four metrics into a single complexity metric. Next, we order the samples in each bin according to the PCA scores. Then, we perform systematic sampling within each bin, using the interval values derived from the sampling size of each bin. The implementation details of our function selection process are provided in §A.3.

## 4 IMPLEMENTATION

In this section, we share a part of the implementation that is crucial to understanding our selection process and experiments. Due to space constraints, we include the implementation details of complexity metrics (§3.2) and function selection method (§3.3) in §A.2 and §A.3, respectively.

### 4.1 PREPARATION OF C CODE FOR TRANSPILATION

**Preprocessing C Files.** We perform preprocessing on C files to resolve macros and dependencies within a codebase. Reducing the input from multiple dependent C files to independent C files is necessary to handle challenges of input/output token limits in LLM-based transpilation. To this end, we modify the build configurations of each codebase to generate preprocessed C files, specifically updating the build files with the `-S` option of GCC, which outputs preprocessed C code instead of compiled binaries. We work on those preprocessed C files in the segmentation process.

**Segmentation of C Files.** We split C files into individual functions due to two main challenges: (1) input/output token limits of LLMs, and (2) while generating compilable and correct output is already difficult for single functions using LLMs, it becomes nearly impossible for multiple functions. Although LLM-based transpilation is relatively new, one study demonstrates the effectiveness of segmentation in C-to-Rust transpilation (Shiraishi & Shinagawa, 2024). However, the proposed approach is not publicly available.

We build an LLVM tool to split a preprocessed C file into individual functions. The tool first identifies the start and end lines of each function in a given C file, and then extracts them into separate C files. The source code of the tool is available in our artifacts.

### 4.2 TRANSPILATION OF C FUNCTIONS INTO RUST

We require a method for transpiling C code into Rust to (1) extract the complexity metrics from the generated Rust code, and (2) obtain feedback from the transpilation process to evaluate the selected functions. To this end, we build a simple transpilation tool with a compilation-error fixing loop that can operate with any local LLM. Our choice of LLM for the experiments is discussed in §5.2.

The tool takes a C file containing an individual function as input and combines it with the instructions shown in Figure 2 of §A.1 before sending a request to the LLM. The LLM performs the transpilation and returns the resulting Rust code in the specified format. The tool then attempts to compile the Rust code. If compilation is successful, the transpiled code is retained, and the process ends with success. If compilation fails, the tool passes the errors from the Rust compiler to the fixing module along with the corresponding Rust code.

The fixing module sends a request to the LLM with the combination of Rust code, compilation errors, and fixing instructions shown in Figure 3 of §A.1. It then attempts to compile the modified Rust code after the LLM fixes the errors. The fixing module continues to attempt to fix the transpiled code until the compilation error fixing attempt limit is reached. If the Rust code remains non-compilable at the limit, the transpilation process ends with failure. Despite this, we retain these samples in our candidate set, as they are evidently challenging for transpilation and still yield complexity metrics.

Table 1: Transpilation performance of 9 LLMs on the microbenchmark set.

| LLM Name | Average Transpilation Time (sec) | Average Compilation Attempt (#) | Transpilation Success (%) |
|---|---|---|---|
| codegeex4:9b | 92.8 | 10.7 | 51.2 |
| codestral:22b | 94.8 | 4.5 | 92.7 |
| gemma2:9b | 104.9 | 12.4 | 47.0 |
| llama3.2:3b | 79.9 | 12.5 | 59.9 |
| llama3.1:8b | 99.5 | 8.9 | 80.5 |
| mistral:7b | 92.5 | 12.4 | 55.0 |
| qwen2.5-coder:7b | 79.7 | 10.0 | 60.1 |
| qwen2.5-coder:14b | **61.0** | 3.3 | 96.2 |
| qwen2.5-coder:32b | 103.6 | **2.6** | **97.6** |

## 5 EXPERIMENTS AND RESULTS

In this section, we first explain the experiment setup in §5.1. Then, we present a preliminary experiment in §5.2 to select an LLM for C-to-Rust transpilation. Next, in §5.3, we present another preliminary experiment to specify the hyperparameters for the selection process. Finally, we present C2RUST-BENCH, a minimized, representative set for evaluating C-to-Rust transpilation in §5.4.

### 5.1 EXPERIMENT SETUP

**Dataset.** We assembled a large dataset by collecting datasets from previous works (Emre et al., 2021; Zhang et al., 2023; Yang et al., 2024; Hong & Ryu, 2024). These works focus on C-to-Rust transpilation or on improving transpiled Rust code. Note that our large dataset also contains c2rust examples (Immunant, 2025b), which are included in the four datasets. There are 64 real-world programs and 1 synthetic program set in our large dataset, all of which come from those 4 previous works. After preprocessing and segmentation steps, we obtained 15,503 functions in total. We give a detailed look at the programs of the large dataset in Table 3 of §B.

**Microbenchmark Set.** We have to specify an LLM to transpile C code into Rust as part of the selection process. However, transpiling the entire large dataset with all candidate LLMs is infeasible due to time and resource constraints. To address this, we sample 10% of the functions in the large dataset and obtain 1,573 functions that form the microbenchmark set for LLM selection.

**Resource and Environment.** We run our experiments on a server with an Intel Xeon Silver 4310 CPU and an NVIDIA A30 GPU. The server runs on a Ubuntu 22.04.4 LTS OS. We implement the Rust parsers using the Rust parsing library Syn. We implement the rest of our method in Python.

### 5.2 EVALUATION OF LLMs ON MICROBENCHMARK SET

Our selection approach benefits from both the C and the corresponding transpiled Rust code. Thus, we need to perform transpilation as part of our selection process. Since transpilation of the large set with 15,503 functions is costly, we specify an LLM to use in transpilation. However, we assess how well the selections obtained using the chosen LLM generalize to other LLMs in §B.2.

**Methodology.** We use the transpilation process described in §4.2 to evaluate LLMs on the microbenchmark set. Candidate LLMs were selected based on three criteria: (1) runnable on a local machine, (2) model size under 24 GB, and (3) reported strong performance on code tasks. After excluding smaller, older or nonconforming models (e.g., those returning natural language or fragmented code despite our instructions), we retained 9 candidates. Each LLM transpiles the microbenchmark set, and we collect metrics to guide LLM selection, including the result of the transpilation, initial transpilation time, number of compilation-fixing attempts, and the total fixing time.

**Results.** In Table 1, we present the results of transpilation performed with 9 LLMs. The total time per sample includes the initial transpilation time, compilation error fixing times, and compilation times. Average time indicates the mean transpilation time across the 1,573 samples in the microbenchmark set. Compilation attempt ranges from 0 to 20, recalling that a value of 0 means the code compiled

successfully after the initial transpilation, and a value of 20 indicates transpilation process ends with compilation failure. The average compilation attempt shows the number of fixing attempts performed on average for the samples in the microbenchmark set. Lastly, transpilation success is the percentage of samples producing a compilable Rust code within 20 fixing attempts.

Among 9 LLMs, `qwen2.5-coder:14b` is fastest, averaging 61 seconds per transpilation followed by `qwen2.5-coder:7b` (79.7) and `llama3.2:3b` (79.9). `qwen2.5-coder:32b` requires the fewest compilation-fix attempts (2.6), ahead of `qwen2.5-coder:14b` (3.3) and `codestral:22b` (4.5). Lastly, `qwen2.5-coder:32b` achieves the highest transpilation success rate (97.6%), surpassing `qwen2.5-coder:14b` (96.2%) and `codestral:22b` (92.7%). We select `qwen2.5-coder:32b` for subsequent experiments due to its top success rate, prioritizing compilation reliability for future benchmark use.

### 5.3 TUNING THE HYPERPARAMETERS

The selection process has two hyperparameters, the number of partitions per dimension and the ratio of sampling per bin. In this section, we present the experiments to identify their optimal values.

**Methodology.** To identify the optimal hyperparameters, we test various combinations of the two hyperparameters in the function selection process described in §A.2 and §A.3. Evaluating the set of selections requires feedback from the target task, transpilation. While we could use transpilation results as binary feedback, it would be limited in assessment of selections. Instead, we leverage compilation-error fixing attempts from transpilation as feedback to assess the representativeness of the selections. Since the selected set is a subset of the large set, the distributions of compilation-error fixing attempts would differ between the two sets. To address this, we normalize the fixing attempt distribution of the large set by the ratio of sample sizes in both sets to obtain the expected distribution.

$$relative\_difference = \frac{1}{21} \cdot \sum_{i=0}^{20} \frac{|expected\_value_i - observed\_value_i|}{expected\_value_i}. \tag{1}$$

We calculate a difference score to justify the representativeness of the function sets selected with different hyperparameters. We use relative difference instead of absolute difference for a fair comparison, since the number of selected functions varies for each hyperparameter combination. We calculate the sum of differences from each compilation error fixing attempt and get the average. The calculation used for each combination of hyperparameters is shown in Equation 1.

For the number of partition hyperparameter, we test 20 values, ranging from 1 to 20. For the ratio of sampling hyperparameter, we test 100 values, ranging from 0.002 to 0.2 by a step size of 0.002. Since these hyperparameters interact, we tune them jointly by combining and testing their candidate values to find a global optimum. These combinations yield 2,000 total selections.

**Results.** In Figure 1, we present the relative differences calculated for 2,000 different selections from the combinations of two hyperparameters. The combinations of the two hyperparameters are on the x-axis in the format of (number of partition, ratio of sampling). Each point on the x-axis corresponds to a unique combination of the two hyperparameters, ordered first by the first hyperparameter, and then by the second. At index 0 of the x-axis (the leftmost in Figure 1), we have the combination (1, 0.002) and at the rightmost, the combination (20, 0.2).

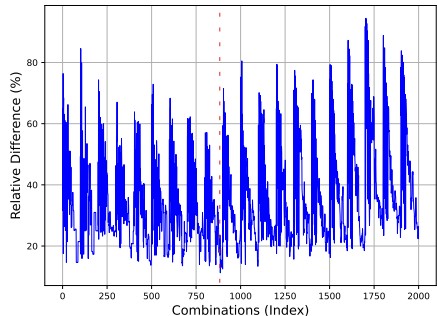

Figure 1: The change of relative difference over combinations of values of two hyperparameters.

To select the best hyperparameters, we aim to minimize the relative difference score. The combination (9, 0.166), located at the index of 883 and marked by the red dashed line in Figure 1, has the lowest relative difference score of 11.2%. Thus, we select 9 and 0.166 as the number of partition and ratio of sampling parameters, respectively.

Table 2: Summary of key C constructs and their occurrence in C2RUST-BENCH.

| C Construct | Total Occurrence (#) | Functions with Occurrence (#) | C Construct | Total Occurrence (#) | Functions with Occurrence (#) |
|---|---|---|---|---|---|
| Pointer Type | 1,553 | 803 | If Statement | 6,123 | 1,562 |
| Array Type | 1,134 | 620 | For Loop | 900 | 526 |
| Struct Type | 764 | 505 | While Loop | 691 | 377 |
| Enum/Union Type | 34 | 22 | Switch Statement | 160 | 125 |
| Function Pointer | 48 | 30 | Goto Statement | 131 | 33 |
| Type Casting | 4,419 | 1,636 | Return Statement | 4,156 | 2,148 |
| Memory Management | 349 | 194 | Break/Continue Statement | 1,443 | 314 |
| Memory Operation | 165 | 119 | | | |

## 5.4 C2RUST-BENCH: A MINIMIZED REPRESENTATIVE SET

In this subsection, we provide details from C2RUST-BENCH. After selecting the LLM and hyper-parameters, we applied our selection process to the large dataset of 15,503 functions. It identified 2,905 functions as the representative set of the large dataset, forming C2RUST-BENCH. Our selection process reduced the number of functions by 81.3%, from 15,503 to 2,905. In practical terms, transpiling the large dataset took 246 hours, whereas C2RUST-BENCH required only 52 hours, a reduction of 78.9%. In addition, the total source lines of code (SLoC) is reduced from 195,926 to 50,150 by 74.4% decrease. We share per-program details from C2RUST-BENCH in Table 4 of §B.

To provide a detailed view of C2RUST-BENCH, we identify the challenging C constructs for C-to-Rust transpilation based on the challenges addressed by prior studies and our own observations. In Table 2, we report the frequency of these constructs in C2RUST-BENCH. The "Total Occurrence" column shows how many times each construct appears across all functions, while the "Functions with Occurrence" column indicates how many of the 2,905 functions contain at least one instance of the construct. With this data, we aim to provide insight for future work leveraging C2RUST-BENCH.

Translating pointers from C to Rust is a major challenge studied in previous works (Emre et al., 2021; Zhang et al., 2023). C2RUST-BENCH contains 1,553 pointer variables across 803 functions, providing a valuable resource for evaluating C-to-Rust transpilers. Arrays and structs also pose challenges due to differences in memory layout, ownership, and mutability semantics. C2RUST-BENCH contains 1,134 array variables across 620 functions and 764 struct variables across 505 functions, offering a comprehensive basis for testing C-to-Rust transpilers.

Several C constructs in C2RUST-BENCH present unique translation challenges due to limited or non-direct equivalents. C2RUST-BENCH contains 34 `union` and `enum` types, requiring careful handling due to C's overlapping memory layouts and Rust's strict type safety, often requiring redesigns with enum variants or unsafe code. Function pointers, appearing 48 times, complicate transpilation due to differences in calling conventions and the need to express dynamic behavior safely in Rust. Finally, type casting, appearing 4,419 times across 1,636 functions, is crucial in C-to-Rust transpilation, as Rust's stricter type system requires many C casts to be rewritten or wrapped in unsafe blocks.

Control flow constructs are fundamental to program semantics, and preserving their behavior during translation is essential. C2RUST-BENCH contains 6,123 `if` statements, 900 `for` loops, and 691 `while` loops, covering a wide range of structured branching and iteration patterns. These constructs test whether transpilers can correctly map C control flow to Rust. Furthermore, 160 `switch` statements and 131 `goto` statements pose challenges, as `switch` maps to Rust's `match`, requiring restructuring, and `goto`, unsupported in Rust, requires extensive rewriting. C2RUST-BENCH also includes 4,156 `return` statements across 2,148 functions and 1,443 `break`/`continue` statements across 314 functions. Prior work (Hong & Ryu, 2024) emphasizes return statements in C-to-Rust transpilation, showing how C output parameters can be replaced with Rust algebraic data types to ensure correct function translation. Together, these features make C2RUST-BENCH an excellent resource for evaluating how well transpilers handle control flow in C-to-Rust translation.

Memory management is a key concern in C-to-Rust transpilation, as C allows manual allocation (`malloc`) and deallocation (`free`), while Rust enforces strict memory safety through its ownership system. C2RUST-BENCH contains 349 memory operations across 194 functions, providing cases to evaluate whether transpilers correctly use ownership, borrowing, and lifetimes to replace manual memory handling. Additionally, it includes 165 memory operations (e.g., `memcpy`, `memset`) across 119 functions, which require translation into safe, idiomatic Rust alternatives such as slice copying or initialization routines. These constructs challenge transpilers to preserve semantics while ensuring memory safety, making them essential for assessing the robustness of C-to-Rust translation tools.

## 6 RELATED WORK

**Transpilation.** C2rust is the most popular rule-based transpilation tool, which produces compilable Rust code from given C code while preserving semantics (Immunant, 2025a). However, Rust code produced by c2rust is covered in unsafe blocks and is not idiomatic. Prior work seeks to improve c2rust output by reducing unsafe usage via compiler feedback (Emre et al., 2021) and applying ownership analysis to convert pointers (Zhang et al., 2023). Another work aims to improve c2rust output by replacing the output parameters with Rust's algebraic data types (Hong & Ryu, 2024).

Despite prior efforts, rule-based transpilation is fundamentally limited in C-to-Rust transpilation. Leveraging recent advances in LLMs for code tasks, the new line of work aims to solve C-to-Rust transpilation using LLMs. VERT combines WebAssembly and LLMs to generate correct and readable Rust code transpiling from various languages, including C (Yang et al., 2024). Moreover, another work aims to overcome the limited context windows of LLMs by segmenting input C code and transpiling smaller units into Rust code (Shiraishi & Shinagawa, 2024).

Concurrent with our work, CRUST-Bench (Khatry et al., 2025) provides a repository-scale benchmark with safe Rust interfaces and test suites for end-to-end evaluation. Unlike our approach, it does not use a systematic selection process or aim to minimize dataset size. C2Rust-Bench instead offers a compact, function-level benchmark built through principled selection, enabling efficient and repeatable evaluation across both rule-based and LLM-based systems. The two benchmarks target different phases of C-to-Rust transpilation evaluation and are therefore complementary.

**Dataset Reduction in ML.** A well-established line of research in ML focuses on instance and coreset selection to reduce the training set. Instance selection studies focus on choosing a subset of representative instances from the original dataset that maintains the overall structure of the data (Olvera-López et al., 2010). They aim to preserve important patterns, such as class distributions and feature relationships, while removing redundancy. Coreset selection seeks a smaller, weighted subset that approximates the distribution of the entire dataset with minimal accuracy loss (Har-Peled & Kushal, 2005; Tsang et al., 2005; Bachem et al., 2017; Novikov et al., 2021).

Recent work has shifted focus to coreset selection for neural networks (Sener & Savarese, 2017; Lee et al., 2024; Song et al., 2025). A previous study redefines active learning for convolutional neural networks as coreset selection, showing that selecting data based on geometric properties outperforms traditional active learning heuristics (Sener & Savarese, 2017). Another work introduces coreset selection for Object Detection (CSOD) and outperforms random selection in object detection tasks (Lee et al., 2024). Lastly, a recent work introduces the SubPIE algorithm that optimizes coreset coverage using entropy-based methods and discrete coordinate descent (Song et al., 2025).

## 7 CONCLUSION

Dealing with memory safety vulnerabilities for more than two decades showed that it is a never ending problem. Thus, migrating from the memory-unsafe C to memory-safe Rust, is seen as the promising solution that can mitigate such vulnerabilities. However, given the scale of C codebases, manual migration is impractical, requiring an automated C-to-Rust transpilation framework. As with other program analysis tasks, a representative dataset is required to evaluate proposed automated transpilers. Such a dataset must be minimized to reduce resource consumption while covering representative samples. In this work, we apply our selection method to reduce the large dataset from previous works, from 15,503 functions to 2,905 functions, which form C2RUST-BENCH. We propose C2RUST-BENCH as a valuable resource for future evaluations of C-to-Rust transpilation tools.

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

## A  IMPLEMENTATION DETAILS

### A.1  TRANSPILATION OF C FUNCTIONS INTO RUST

In Figure 2, we present the instructions sent to LLM along with the input C function for the initial transpilation from the C to Rust.

---

Behave like you are an expert of C and Rust. Behave like you are a translator from C language to Rust language. Can you translate C code given above into Rust code?
Do not explain the code to me! Only return Rust code correspoding to the given C code.

Follow these intructions strictly in translation:
(1) Do not add any extra error handling,
(2) Do not merge functions,
(3) Do not change variable names,
(4) Use no_mangle for each function,
(5) Make each function public,
(6) Translate the standard C library function calls by placing a decoy function call (leave the decoy function body empty if possible) with the same name, and
(7) Only return a Rust code and nothing else!

---

Figure 2: The instructions given to LLM for initial transpilation.

In Figure 3, we present the instructions sent to LLM along with the compilation errors to fix the compilation errors previously obtained from the compilation attempt of transpiled Rust code.

> When attempted to compile the recently generated rust code, I obtained the compilation errors given above. Fix those errors and only return the modified Rust code. Do not explain the code or changes to me!

Figure 3: The instructions given to LLM for fixing compilation errors.

## A.2 DATA COLLECTION OF CODE METRICS

In this subsection, we present how we implement the data collection of the code complexity metrics. The details of how we utilize existing tools and build parsers to get complexity metrics can be found in the GitHub repository [3].

**Maintainability Index** There are tools to measure Maintainability Index (MI) of a code from C and Rust languages. We use the open-source tool published on github to obtain MI metric for a C function Jarod42 (2025). We employ the rust code analysis tool published in a previous work to obtain MI metric for the corresponding Rust code Ardito et al. (2020).

**Unsafe Operation Complexity** We implement unsafe operation complexity metric by building a parser in Rust, specifically designed to identify and extract unsafe blocks from a given Rust function. One important detail is that we choose to count unsafe blocks, as well as statements inside the blocks, because counting only blocks would be limited to express the complexity of unsafe operations, as unsafe blocks may contain a variable number of statements, ranging from a single operation to a large number of operations.

As a result, the parser produces two key outputs: the total number of unsafe blocks and the set of number of statements contained in each block. To represent unsafe operations as a unified metric, we calculate the average number of unsafe statements per unsafe block. Using this metric, we capture not only the frequency of unsafe code usage but also its density. The lower boundary of this metric is zero, indicating the absence of unsafe code, while the upper boundary is unbounded, reflecting the potential for increasingly complex unsafe operations.

**Data Type Complexity** In Rust, variables can be explicitly typed, or the type may be inferred by the compiler based on the value assigned. To quantify the complexity introduced by data types, we build a parser to collect and analyze the types of variables within a given Rust code. The parser extracts the type of a variable when it is explicitly specified in the declaration. For example, when a variable is defined as `let x:i32=5`, the parser identifies the type as `i32` for the variable `x`.

However, unlike languages such as C, Rust allows variables to be defined without explicitly stating their type. In these cases, the parser examines the right-hand side of the declaration, if available, and infers the variable's type based on the right-hand side expression. For example, in a declaration such as `let y=10`, the parser infers that the type of variable `y` is `i32`, since the value of `10` is an integer literal.

After extracting all the variable types, we get the set of unique types and obtain the total number of types in the set. We use the number of unique types as a metric to represent the complexity of the data types for a given Rust function.

## A.3 FUNCTION SELECTION USING PARTITIONING

As a first step in obtaining bins, we divide each of the dimensions, corresponding to the complexity metrics, into partitions based on the value of the width specified by the following formula:

$$width = \frac{max\_value - min\_value}{number\_of\_partition} \tag{2}$$

The $number\_of\_partition$ used in Equation 2 is a hyperparameter common for all dimensions, specifying the number of pieces to cut. We present a preliminary experiment identifying the optimal

---

[3]https://github.com/anonymous8428/C2Rust-Bench

Table 3: The detailed information of the programs in the large dataset.

| Name | Function (#) | SloC (#) | Name | Function (#) | SloC (#) |
|---|---|---|---|---|---|
| transcoder-set | 4,012 | 17,684 | hello-2.12.1 | 77 | 979 |
| libxml2 | 1,964 | 26,729 | ed-1.19 | 69 | 691 |
| gprolog-1.5.0 | 758 | 8,119 | brotli-1.0.9 | 62 | 1,084 |
| nettle-3.9.1 | 646 | 8,103 | lodepng | 59 | 542 |
| json.h | 644 | 18,415 | pexec-1.0rc8 | 54 | 598 |
| tmux | 607 | 7,625 | diffutils-3.10 | 45 | 1,593 |
| tulipindicators-0.9.1 | 516 | 5,448 | lil | 40 | 312 |
| libosip2-5.3.1 | 515 | 5,005 | buffer-0.4.0 | 39 | 332 |
| tar-1.34 | 446 | 6,528 | grep-3.11 | 38 | 1,263 |
| less-633 | 343 | 4,947 | libzahl-1.0 | 35 | 653 |
| nano-7.2 | 324 | 4,368 | indent-2.2.13 | 32 | 694 |
| optipng-0.7.8 | 311 | 5,548 | gzip-1.12 | 28 | 780 |
| gawk-5.2.2 | 307 | 7,135 | bzip2 | 22 | 625 |
| mcsim-6.2.0 | 260 | 4,208 | genann | 22 | 317 |
| heman | 257 | 3,179 | snudown | 20 | 293 |
| screen-4.9.0 | 229 | 5,680 | libcsv | 20 | 180 |
| wget-1.21.4 | 226 | 3,338 | quadtree-0.1.0 | 20 | 160 |
| tinycc | 207 | 2,666 | which-2.21 | 18 | 458 |
| patch-2.7.6 | 182 | 2,360 | sed-4.9 | 18 | 184 |
| cflow-1.7 | 181 | 2,604 | urlparser | 18 | 178 |
| mtools-4.0.43 | 172 | 2,118 | robotfindskitten | 12 | 158 |
| make-4.4.1 | 168 | 3,597 | avl | 8 | 62 |
| json-c | 154 | 2,201 | rgba | 7 | 69 |
| rcs-5.10.1 | 154 | 1,955 | bst | 5 | 59 |
| bc-1.07.1 | 153 | 2,581 | xzoom | 4 | 419 |
| uucp-1.07 | 152 | 4,863 | ht | 4 | 38 |
| findutils-4.9.0 | 151 | 2,985 | qsort | 3 | 27 |
| pth-2.0.7 | 137 | 2,016 | libtool-2.4.7 | 2 | 37 |
| dap-3.10 | 129 | 3,942 | grabc | 1 | 48 |
| cpio-2.14 | 118 | 1,536 | libtree-3.1.1 | 1 | 11 |
| binn-3.0 | 116 | 961 | glpk-5.0 | 1 | 8 |
| units-2.22 | 104 | 2,307 | time-1.9 | 1 | 16 |
| enscript-1.6.6 | 78 | 2,273 | | | |

value of $number\_of\_partition$ in §5.3. The width value in Equation 2 is calculated for each dimension, since the metrics have different minimum and maximum values. Based on the width values, each dimension is partitioned into equal pieces. Then, the combination of the partitions in multidimensional space forms the bins. If $number\_of\_partition$ is set to $n$, the number of bins created is equal to $n^4$. However, some of those bins remain empty as anticipated, and we proceed with the non-empty bins.

We select samples from each bin, using Principal Component Analysis (PCA) and systematic sampling. We first calculate a summarized complexity metric from 4 metrics for each sample using PCA. Then, we order samples in each bin using the unified PCA complexity metric. Next, we perform selection from each bin based on the systematic sampling approach, which means that we select from ordered samples that are placed away from each other by an interval. In the following formula, we obtain the interval value to use in systematic sampling:

$$interval = \frac{bin\_population}{bin\_population \times ratio\_of\_sampling} \tag{3}$$

The $ratio\_of\_sampling$ used in Equation 3 is another hyperparameter common for all bins, which specifies the percentage of sample to take from each bin. The optimal value of $ratio\_of\_sampling$ parameter is specified by a preliminary experiment in §5.3. In the denominator of this formula, we have the sampling size calculated by multiplying $subset\_population$ by $ratio\_of\_sampling$, which is different for each bin. We obtain the interval value for each bin by dividing the population size by the sampling size of the bin. Lastly, we select samples from each bin that are placed at

positions separated by the interval value. Consequently, we obtain diverse samples from distinct data points depicted by the bins in multidimensional space and the different positions in each bin.

## B EVALUATION DETAILS

### B.1 DATASET.

In Table 3, we share the details of our large set including functions from 64 real world and 1 synthesized programs. Under the function column, we share the total number of individual functions obtained for each program after preprocessing and segmentation. Under the SLoC column, we share the total number of source line of code for the corresponding functions of each program.

### B.2 CROSS-LLM EVALUATION OF SELECTED FUNCTIONS ON MICROBENCHMARK SET

In this subsection, we perform function selection using the LLM and the hyperparameters previously chosen. Then, we evaluate the selected function set on all 9 LLMs to demonstrate that the selections are generalized to other LLMs.

**Methodology.** We perform function selection from microbenchmark set by setting the LLM, the number of partition per dimension, and the ratio of sampling per bin as `qwen2.5-coder:32b`, 9, and 0.166 respectively. To evaluate the selected functions, we use the compilation error fixing attempt from the transpilation process as in subsection 5.3. First, we obtain the distribution of compilation error fixing attempts for both the selected set and the microbencmark set. We normalize the distributions of the microbenchmark set by the ratio of the two sets. Then, we calculate a relative difference score using the formula shown in subsection 5.3.

As presented in §5.2, the microbencmark set is transpiled using 9 different LLMs. Thus, each of the LLMs has their own compilation error fixing attempt feedback. We calculate the relative difference score for 9 LLMs using their own distributions for selected function set and microbenchmark set. Using their individual feedback allows us to justify that the functions selected using `qwen2.5-coder:32b` are also representative for the other LLMs.

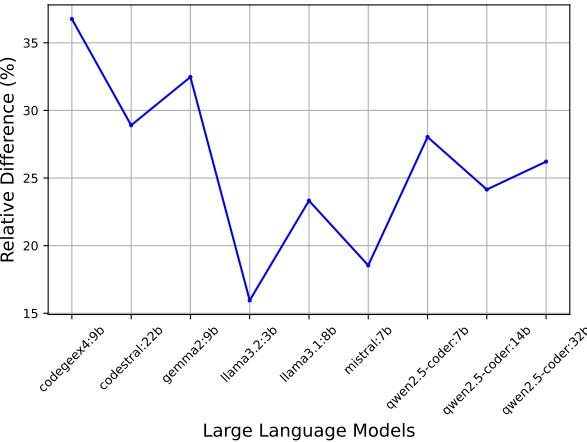

Figure 4: The relative difference for 9 LLMs.

**Results.** In Figure 4, we present the relative difference scores for 9 LLMs. For more details, we share the expected and observed distributions for each LLM in Figure 5. Furthermore, the difference score for `qwen2.5-coder:32b` is different from that presented in subsection 5.3, since this experiment is performed on the microbenchmark set. However, the difference scores of the LLMs are comparable to each other, since we use the same set of hyperparameters for all of them.

The three models of `qwen2.5-coder`, `llama3.1:8b`, and `codestral:22b` achieve reasonably close difference scores. The difference scores of `codegeex4:9b` and `gemma2:9b` are slightly

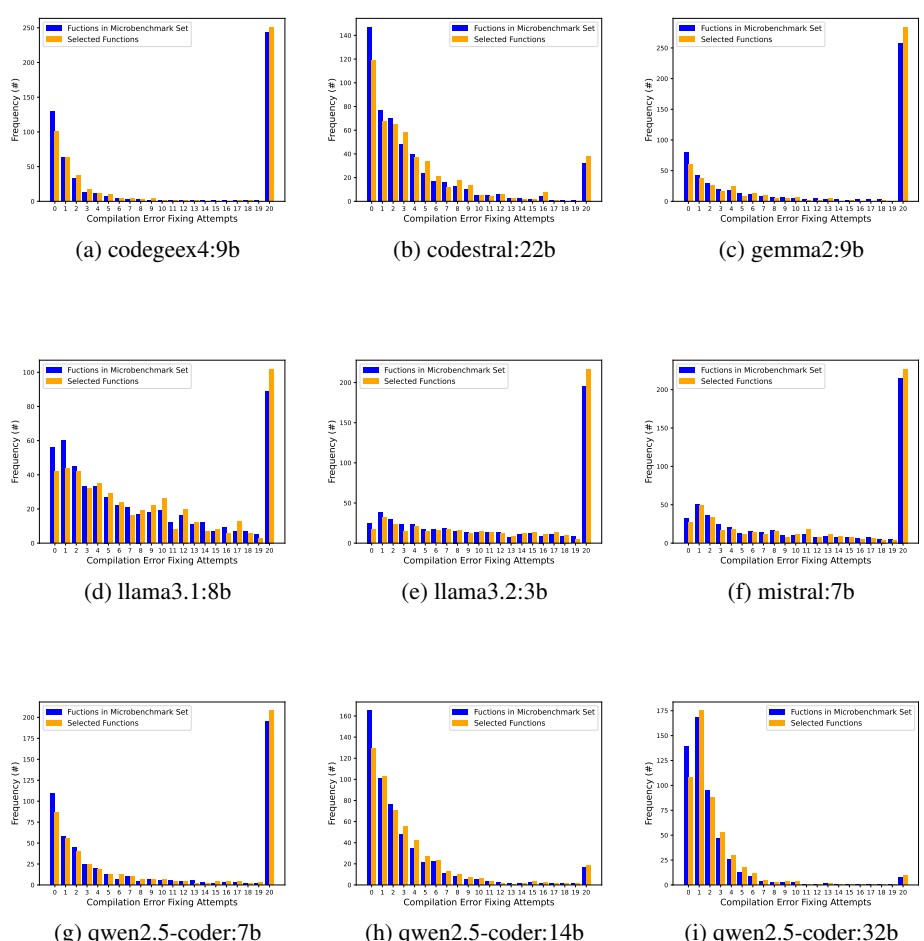

(a) codegeex4:9b  (b) codestral:22b  (c) gemma2:9b

(d) llama3.1:8b  (e) llama3.2:3b  (f) mistral:7b

(g) qwen2.5-coder:7b  (h) qwen2.5-coder:14b  (i) qwen2.5-coder:32b

Figure 5: Compilation error fixing attempt distribution of selected and microbenchmark sets for 9 LLMs.

higher than the previously mentioned 5 LLMs while `llama3.2:3b` and `mistral:7b` are lower than them. Note that the poor performance of `llama3.2:3b` and `mistral:7b` in transpilation leads to uneven distributions with some accumulations as presented in Figure 5, which is the main reason for lower difference scores than `qwen2.5-coder:32b`. Consequently, even though function selection is performed using the transpilation output of `qwen2.5-coder:32b`, the relative difference scores of other LLMs show that the selected functions are representative for all LLMs. Thus, employing a specific LLM as part of the selection process for the C-to-Rust transpilation evaluation dataset does not pose a threat to the validity of our study.

## B.3 SELECTED FUNCTION SET.

In Table 4, we share a detailed look at C2RUST-BENCH. Under the function column, we share the total number of individual functions from each program existing in C2RUST-BENCH. Under the SLoC column, we share the total number of source lines of code for the corresponding functions of each program.

Table 4: The detailed information of the programs in C2RUST-BENCH.

| Name | Function (#) | SloC (#) | Name | Function (#) | SloC (#) |
| --- | --- | --- | --- | --- | --- |
| transcoder-set | 714 | 3,457 | pth-2.0.7 | 23 | 374 |
| libxml2 | 392 | 5,952 | cpio-2.14 | 22 | 293 |
| json.h | 135 | 4,641 | json-c | 22 | 227 |
| gprolog-1.5.0 | 125 | 1,358 | brotli-1.0.9 | 14 | 516 |
| nettle-3.9.1 | 112 | 1,612 | grep-3.11 | 12 | 871 |
| libosip2-5.3.1 | 105 | 1,466 | ed-1.19 | 12 | 126 |
| tmux | 97 | 1,116 | indent-2.2.13 | 11 | 382 |
| tulipindicators-0.9.1 | 93 | 1,071 | hello-2.12.1 | 11 | 125 |
| tar-1.34 | 85 | 1,489 | enscript-1.6.6 | 10 | 1,155 |
| optipng-0.7.8 | 68 | 1,535 | lodepng | 10 | 211 |
| less-633 | 67 | 988 | diffutils-3.10 | 10 | 186 |
| nano-7.2 | 59 | 868 | pexec-1.0rc8 | 9 | 90 |
| gawk-5.2.2 | 56 | 3,182 | gzip-1.12 | 7 | 221 |
| mcsim-6.2.0 | 46 | 759 | lil | 7 | 58 |
| wget-1.21.4 | 46 | 719 | buffer-0.4.0 | 7 | 56 |
| uucp-1.07 | 43 | 2,406 | bzip2 | 5 | 272 |
| heman | 43 | 599 | libzahl-1.0 | 5 | 110 |
| screen-4.9.0 | 42 | 2,066 | libcsv | 5 | 29 |
| tinycc | 42 | 656 | snudown | 4 | 45 |
| mtools-4.0.43 | 42 | 463 | genann | 3 | 84 |
| dap-3.10 | 38 | 1,568 | quadtree-0.1.0 | 3 | 24 |
| cflow-1.7 | 33 | 383 | xzoom | 2 | 380 |
| bc-1.07.1 | 32 | 981 | which-2.21 | 2 | 222 |
| patch-2.7.6 | 31 | 410 | sed-4.9 | 2 | 38 |
| make-4.4.1 | 29 | 1,537 | urlparser | 2 | 27 |
| rcs-5.10.1 | 29 | 422 | avl | 2 | 16 |
| findutils-4.9.0 | 27 | 1,141 | robotfindskitten | 2 | 7 |
| binn-3.0 | 26 | 275 | ht | 1 | 14 |
| units-2.22 | 23 | 786 | | | |

