# OpenReview forum: "C2Rust-Bench: A Minimized, Representative Benchmark for C-to-Rust Transpilation"
_ICLR.cc/2026/Conference — Submitted to ICLR 2026_

### Official Review · Reviewer_xyiR · 2025-10-26

**Soundness:** 3
**Presentation:** 3
**Contribution:** 2
**Rating:** 6
**Confidence:** 3

**Summary:**

This paper introduces C2Rust-Bench, a dataset aimed at providing a minimized and representative benchmark for evaluating C-to-Rust transpilation tools. The authors propose a method to select 2,905 functions from 15,503 C functions from existing literature. The method is based on four code complexity metrics, partitioning the metric space, and selecting functions via systematic sampling guided by Principal Component Analysis. The authors claim that this benchmark retains the essential characteristics of the original set while drastically reducing evaluation time.

**Strengths:**

1.Clear Problem Identification: The paper correctly identifies a significant pain point in the emerging field of C-to-Rust transpilation: the lack of a standard, manageable benchmark. The motivation, supported by recent reports, is strong and relevant to the community.
2.Practical Utility: The achieved reduction in dataset size (81.3%) and the corresponding decrease in evaluation time (≈80%) are substantial and directly address the problem of lengthy experiments. Releasing the dataset and code is a positive practice that facilitates adoption and reproducibility.
3.Cross-LLM Validation: The experiment in Appendix B.2, which tests the representativeness of the selected set across different LLMs, is a thoughtful addition. It helps mitigate the concern that the benchmark is overly tailored to the specific LLM (qwen2.5-coder:32b) used in its creation.

**Weaknesses:**

1.Fundamental Circularity in Methodology: A core weakness lies in the methodology's reliance on a specific LLM's transpilation output. The Rust-side metrics (MI-Rust, Unsafe Complexity, Data Type Complexity) are derived from code generated by qwen2.5-coder:32b. If this LLM has systematic biases or errors in how it handles certain C constructs (e.g., pointers, complex types), these errors are baked into the complexity metrics used for selection. This creates a potential feedback loop where the benchmark is "representative" of what a specific LLM finds challenging, not necessarily of the intrinsic challenges of C-to-Rust transpilation. While the cross-LLM experiment provides some reassurance, it does not fully resolve this conceptual issue.
2.Validation via Proxy is Incomplete: The primary validation uses the distribution of compilation-error fixing attempts as a proxy for representativeness. While a useful signal, this is a narrow measure of transpilation quality. It does not account for:
*Semantic Equivalence: A compilable Rust function can be semantically incorrect.
*Idiomaticity: The benchmark does not measure how "Rust-like" the output is, a key concern in prior work.
*Runtime Behavior: Correct compilation does not guarantee correct execution. A more robust validation would involve a sample-based check for semantic correctness, perhaps using existing test suites from the original C programs.
3.Lack of a Strong Baseline for Comparison: The paper does not compare its selection method against a strong baseline, such as random sampling with the same reduction ratio. It is plausible that a carefully stratified random sample could achieve a similar level of "representativeness" in terms of code construct coverage and compilation attempt distribution, with a much simpler methodology. The added value of the complex PCA-and-binning approach over simpler methods is not conclusively demonstrated.
4.Limited Scope of "Representativeness": The benchmark is built at the function level. While this is justified by LLM context limits, it ignores critical challenges in transpiling programs, such as inter-procedural analysis, global state management, and module/system architecture translation. The benchmark is thus primarily useful for evaluating function-level translation engines (like LLMs) and may be less applicable for evaluating tools that operate on a whole-program level.

**Questions:**

1.Given the circularity concern, did the authors consider using the output of a rule-based transpiler (like c2rust) to calculate the Rust-side metrics, as it would provide a more deterministic and predictable baseline?
2.How does the performance of a simple random sampling baseline compare to the proposed method on the same representativeness metric?
3.The hyperparameter tuning uses the same proxy metric (compilation attempts) for optimization. Could this lead to overfitting to this specific, imperfect measure of quality?

---

> ### Author Response · Authors · 2025-11-20
>
> **1. Fundamental circularity in methodology**
>
> We acknowledge the concern regarding the potential bias. To mitigate this, we plan to revise our approach in the next iteration of the paper. Specifically, we will calculate the Rust-side metrics, such as the Maintainability Index and unsafe code complexity, from the output of rule-based transpilers like c2rust, which will provide a more deterministic and predictable baseline for these metrics as suggested. This adjustment will help reduce model-dependent bias and align the metrics more closely with standard practice in C-to-Rust transpilation.
>
> **2. Validation via proxy is incomplete**
>
> We appreciate the reviewer’s concern that validation based solely on compilation outcomes does not capture semantic correctness, idiomaticity, or runtime behavior. We agree these aspects are important, but in our work compilation outcomes serve only to verify that the selected subset preserves the difficulty distribution of the full 15,503-function pool. Compilation success is not used to measure model correctness or semantic fidelity; it is a proxy to ensure that the reduced dataset spans the same easy, medium, and hard regions as the original collection. Thus, compilation attempts do not affect benchmark scoring or LLM evaluation, but confirm that the minimized benchmark preserves the structural difficulty landscape.
>
> **Idiomaticity**
>
> While we do not directly measure “Rust-like” idiomaticity, the Maintainability Index (MI) indirectly captures code quality, readability, and complexity, which correlate with idiomatic Rust. Using MI in selection ensures that the benchmark reflects function-level challenges relevant to both correctness and producing structured, maintainable code.
>
> **Semantic correctness and runtime validation**
>
> We recognize that semantic correctness and runtime validation are closely related concerns and face similar fundamental limitations. Accordingly, we address them together here, as we have already discussed these constraints in our response to Reviewer fNEx. The challenges to runtime validation are as follows:
>
> 1. **Executing transpiled Rust functions is infeasible.**
>    Function-level translation lacks module structure, type definitions, global state, and lifetime/aliasing context. These cannot be reconstructed automatically at scale, making execution of the transpiled Rust functions impossible, even though the original C programs are runnable.
>
> 2. **No unit tests exist for most functions.**
>    The source projects do not provide per-function tests, and generating thousands of them is infeasible and would introduce bias. Many functions also rely on non-local invariants that cannot be synthesized.
>
> 3. **Undefined behavior breaks semantic comparison.**
>    Many functions rely on C undefined behavior, where Rust semantics differ. Execution tests would produce inconsistent and misleading results.
>
> 4. **Execution filtering would bias the benchmark toward trivial cases.**
>    Only simple, dependency-free functions could be tested, removing exactly the pointer-heavy and unsafe constructs needed to measure C-to-Rust difficulty.
>
> 5. **Compilation difficulty is aligned with structural translation difficulty.**
>    Borrow-check failures, type mismatches, lifetime errors, unsafe misuse, and pointer issues map directly to the complexity metrics we track, making compilation attempts an appropriate proxy for difficulty only for dataset validation, not evaluation.
>
>
> **3. Lack of a strong baseline for comparison**
>
> We acknowledge the importance of comparing against a baseline. While random sampling could produce a subset, our complexity-based stratified sampling preserves key C code dimensions, including control flow, pointer operations, and unsafe usage, which are critical for measuring C-to-Rust difficulty. In the revised version, we will compare our method to random sampling at the same reduction ratio to demonstrate its effectiveness in maintaining structural diversity and difficulty distribution, showing the benefits of complexity-based selection.
>
> **4. Limited Scope of Representativeness**
>
> We acknowledge that C2Rust-Bench’s function-level scope does not capture full-program challenges such as inter-procedural analysis, global state management, or module-level architecture. It is designed as a micro-benchmark for rapid, controlled evaluation of individual C functions, providing a repeatable method to measure function-level transpilation performance. While whole-program realism is important, C2Rust-Bench complements benchmarks like CRUST-Bench, which address repository-level evaluation and end-to-end correctness. Its focus on memory management, pointer arithmetic, type casting, and control flow enables iterative testing and comparison of transpilers at a granular level. The revised version will clarify that C2Rust-Bench is intended to support function-level insights, helping researchers refine tools before scaling to complete programs.

---

### Official Review · Reviewer_n7AF · 2025-10-29

**Soundness:** 3
**Presentation:** 3
**Contribution:** 2
**Rating:** 6
**Confidence:** 4

**Summary:**

C2RUST-BENCH is a compact, function-level benchmark designed to evaluate C2Rust transpilation with high coverage and low redundancy. Starting from a large corpus of real-world C code, the authors preprocess and split files into individual functions, then select 2.9k representative items from ~15k candidates. Selection is driven by multiple complexity signals: Maintainability Index in C and in transpiled Rust, the presence and extent of unsafe features, and data-type/operation complexity (e.g., pointers, casts, arrays/structs, control flow). They stratify these metrics into bins, compute an aggregate complexity score (via PCA), and systematically sample across easy-to-hard regions to avoid a mid-difficulty bias. Difficulty is further grounded by empirical transpilation feedback: functions are passed through an LLM-based pipeline to observe compile success and the number of fix attempts, and both successes and failures remain in the set. The resulting benchmark stresses known pain points—pointer arithmetic, type casts, function pointers, goto/switch—while remaining small enough for rapid iteration, ablation studies, and leaderboard-style comparisons. Positioned as complementary to repository-scale suites (e.g., CRUST-Bench), C2RUST-BENCH prioritizes controlled, repeatable micro-evaluations over end-to-end realism. It’s most suitable for measuring incremental improvements in transpiler reliability, error modes, and code-quality trade-offs, and for probing how specific C constructs affect Rust compilation and safety outcomes.

**Strengths:**

* Principled, balanced curation: Uses multi-metric stratified sampling (complexity, unsafe usage, types) to cover a broad easy to hard spectrum rather than skewing to median cases.

* Empirical difficulty grounding: Calibrates “hardness” with real transpilation/compile outcomes (incl. number of fixes), not just static code metrics—improves relevance for evaluation.

**Weaknesses:**

1. Limited ecosystem realism: Function-level snippets miss repo-level issues (build systems, headers/linking, cross-file types, macros), so results may not transfer to end-to-end C to Rust.

2, Potential measurement bias: Difficulty signals depend on one/few transpilers and chosen metrics/PCA; different tools or metrics could shift what’s labeled “hard,” affecting generality.

**Questions:**

What is the fundamental challenge to translate C to Rust? How your approach is designed towards this challenge?

---

> ### Author Response · Authors · 2025-11-20
>
> **1. Limited ecosystem realism**
>
> We acknowledge that function-level snippets may miss repo-level issues. C2Rust-Bench is explicitly designed as a function-level micro-benchmark for rapid, iterative evaluation of transpilation systems, focusing on individual C functions rather than complete programs. Our intention is to provide a controlled, repeatable means of measuring transpilation performance, not to replicate the full ecosystem complexity found in larger codebases.
>
> While end-to-end realism is indeed important for evaluating full-scale transpilers, we see C2Rust-Bench as complementing benchmarks like CRUST-Bench (Khatry et al.), which do cover repository-level concerns and include manual test suites for end-to-end correctness. The primary goal of C2Rust-Bench is to provide a focused benchmark that allows for micro-evaluations of transpilation tools, specifically targeting function-level challenges such as memory management, pointer arithmetic, type casting, and control flow. We believe that our benchmark offers a valuable contribution to the field by enabling rapid iterations and comparison across different transpilers in a controlled environment.
>
> In the revised version, we plan to clarify the purpose of our dataset, explicitly highlighting that it is intended to be complementary to other benchmarks like CRUST-Bench, which cover full program transpilation and end-to-end validation. C2Rust-Bench aims to provide insights into function-level challenges, helping researchers refine and evaluate transpilation tools that operate at a granular level before scaling up to complete programs.
>
> **2. Potential measurement bias**
>
> We acknowledge the concern regarding potential measurement bias, particularly the possibility that difficulty signals may depend on the chosen transpilers and metrics. While it is true that different tools or metrics could influence what is labeled as “hard,” we believe that the complexity metrics we have selected, such as control flow, unsafe code usage, and data type complexity, capture a broad spectrum of challenges inherent to C-to-Rust transpilation. These metrics are grounded in fundamental aspects of the C and Rust languages, such as pointer manipulation, type casting, and memory safety, which are unlikely to vary drastically across different transpilers.
>
> Additionally, while different transpilers may exhibit varying success rates or performance characteristics, the relative difficulty of individual functions (i.e., which functions are easy, moderate, or hard) is likely to remain consistent. The structural features of the C code, such as the presence of complex data structures or memory operations, serve as a strong predictor of transpilation difficulty, independent of the specific tool used.
>
> Thus, although different tools or metrics could introduce some variability in the absolute difficulty scoring, we are confident that the comparative difficulty across functions would remain stable. This means that the selection of representative functions would not be significantly impacted by the choice of transpiler or model, preserving the integrity and generalizability of the benchmark across different tools. Therefore, while the possibility of measurement bias exists, it does not compromise the validity or usefulness of our approach.

---

### Official Review · Reviewer_LffA · 2025-10-30

**Soundness:** 3
**Presentation:** 2
**Contribution:** 3
**Rating:** 6
**Confidence:** 3

**Summary:**

This paper introduces C2Rust-Bench, a minimized but representative benchmark dataset for evaluating C-to-Rust transpilation systems. The authors first argue that existing evaluations rely on large or manually curated datasets that hinder reproducibility and efficiency. Then they observe that there is an upper bound beyond which increasing program size no longer contributes to the complexity or representativeness of transpilation. To address these issues, the authors propose a principled reduction method that extracts C functions from prior datasets using quantitative complexity metrics, including the Maintainability Index (for both C and Rust), unsafe operation density, and data type diversity. A PCA-based scoring and sampling algorithm is used to preserve representativeness and diversity. The benchmark is validated through cross-LLM transpilation experiments on nine different models. Both the dataset and source code are publicly released.

**Strengths:**

1. Standardizing evaluation in the growing field of AI-assisted code transpilation from C to Rust is timely and well-motivated.
2. The dataset construction uses several types of complexity metrics with PCA-based sampling, which is technically sound and interpretable.
3. The empirical validation across nine LLMs demonstrates generality and robustness.
4. Public release of both the dataset and code significantly enhances reproducibility and potential community impact.

**Weaknesses:**

1. Several metrics (e.g., Maintainability Index and unsafe complexity) are computed on transpiled Rust code, not on ground-truth Rust implementations, which may introduce model-dependent bias.
2. The correctness validation focuses only on compilation success, which does not ensure semantic equivalence between the C and Rust code. More rigorous behavioral or semantic checks would strengthen the claims.
3. The representativeness is guaranteed by minimizing the relative difference score, which is somewhat heuristic. It does not directly show that the selected subset of functions preserves semantic or structural diversity.

**Questions:**

1. Have you verified whether the selected functions in C2RUST-BENCH cover similar functional semantics or API categories as those in the full dataset?
2. Since both MI and unsafe complexity depend on the LLM-generated Rust code, how do you ensure these metrics reflect the intrinsic complexity of the original C functions rather than translation artifacts introduced by the LLM?

---

> ### Author Response · Authors · 2025-11-20
>
> **1.Metrics computed on transpiled Rust code**
>
> We acknowledge the concern regarding the potential bias introduced by using metrics derived from LLM-generated Rust code. To mitigate this, we plan to revise our approach in the next iteration of the paper. Specifically, we will calculate the Rust-side metrics, such as the Maintainability Index and unsafe code complexity, based on the output of rule-based transpilers like c2rust, which will provide a more deterministic and predictable baseline for these metrics. This adjustment will help reduce model-dependent bias and align the metrics more closely with standard practice in C-to-Rust transpilation.
>
> **2. Compilation validation and semantic correctness**
>
> We appreciate the reviewer’s concern regarding semantic correctness, which is similar to the concern shared by Reviewer fNEx. While we fully agree that semantic correctness is essential when evaluating a transpiler, the focus of our work is on compilation outcomes as a validation signal to ensure that the selected subset of functions preserves the same difficulty distribution observed in the full 15,503 function pool. In other words, compilation success is not used as a metric of model correctness; it is only a proxy to confirm that the reduced dataset spans the full range of difficulties, ensuring that the selected functions are representative of the original collection. Therefore, compilation attempts are not a critical part of benchmark scoring or assessing LLM performance, but only serve to verify that the benchmark maintains the structural difficulty landscape of the larger corpus.
>
> While we agree that introducing execution-based correctness could help confirm that the compilation-based validation is not masking significant semantic issues, execution-level validation at the function granularity is difficult for several fundamental reasons:
>
> 1. **Executing transpiled Rust functions is infeasible.**
>    Function-level translation lacks module structure, type definitions, global state, and lifetime/aliasing context. These cannot be reconstructed automatically at scale, making execution of the transpiled Rust functions impossible, even though the original C programs are runnable.
>
> 2. **No unit tests exist for most functions.**
>    The source projects do not provide per-function tests, and generating thousands of them is infeasible and would introduce bias. Many functions also rely on non-local invariants that cannot be synthesized.
>
> 3. **Undefined behavior breaks semantic comparison.**
>    Many functions rely on C undefined behavior, where Rust semantics differ. Execution tests would produce inconsistent and misleading results.
>
> 4. **Execution filtering would bias the benchmark toward trivial cases.**
>    Only simple, dependency-free functions could be tested, removing exactly the pointer-heavy and unsafe constructs needed to measure C-to-Rust difficulty.
>
> 5. **Compilation difficulty is aligned with structural translation difficulty.**
>    Borrow-check failures, type mismatches, lifetime errors, unsafe misuse, and pointer issues map directly to the complexity metrics we track, making compilation attempts an appropriate proxy for difficulty only for dataset validation, not evaluation.
>
> **3. Representativeness and structural diversity**
>
> We agree that the relative difference score is a heuristic and does not directly validate semantic or structural diversity. However, it serves as a useful metric for ensuring that the selected subset of functions spans a range of complexities.
>
> To further support our claim of representativeness, we will enhance the current submission by including an additional table that provides detailed statistics on key C constructs (such as pointers, type casting, and memory management) and their occurrences in the large set, alongside the minimized C2Rust-Bench. This will allow us to demonstrate how well the selected subset preserves the full spectrum of C constructs and ensures that the benchmark maintains its broad applicability to various transpilers. While Table 2 of the original submission already provides this data for the minimized benchmark, this new table will focus on the large set, further validating that our selection process preserves both semantic and structural diversity of the large set.

---

### Official Review · Reviewer_fNEx · 2025-11-01

**Soundness:** 2
**Presentation:** 2
**Contribution:** 1
**Rating:** 2
**Confidence:** 5

**Summary:**

The paper proposes a funtion level benchmark towards measuring C to Rust transpilation.  The paper proposes curation of a minimalized set of tasks that are selected using 3 metrics, the maintainability index, unsafe code complexity, and data type complexity. The authors propose a pipeline towards generating a benchmark that measures transpilation of language models at the function level. The final benchmark contains various constructs like deal with memory management, control flow constructs, etc.

**Strengths:**

The paper proposes a benchmark, with an LLVM tool that can automatically create large amounts of function-level data. The final dataset covers various aspects like pointer types, type casting, memory management, control flow constructs, etc. The authors use pre-existing work to curate a large set of functions using an automated technique

**Weaknesses:**

- The benchmark measures success using compilation success but not execution-based correctness (unit test/integration tests). Not having validation correctness is a __severe limitation__ of the current work, given that a trivial implementation could also pass the current success measure.
- Limited evaluation on models, why are closed-source models not accounted in the evaluation? Is it too trivial for LLMs like gpt-5, claude-4.5-sonnet to transpile function-level C code, given that qwen-2.5-coder gets ~97% (after 3 rounds of repair)?
- The authors claim that the benchmark is a representative set, but do not perform an analysis of various kinds of errors made by language models (unsafe, borrowing, type mismatch, etc.) when transpiling from C to Rust.
- The paper makes a strong claim on line 244 that generating compilable and correct output by an LLM is nearly impossible for multiple functions, while this is not the case, as shown in [CRUST-bench (Khatry et. al.)](https://arxiv.org/abs/2504.15254) for closed-source models like o3, gpt-4o, etc.

**Questions:**

Please refer to weaknesses.

Other questions:
- What are the average lines of code in each instance of C2RustBench
- Avg number of arguments, and argument types?

Minor errors in presentation :
- Line 211 states 4 metrics instead of 3

---

> ### Author Response · Authors · 2025-11-20
>
> **1. Compilation vs. execution-based correctness**
>
> We appreciate the reviewer’s concern regarding execution-based correctness. We fully agree that semantic correctness is essential when evaluating a transpiler. However, in our work compilation outcomes serve a very different purpose: they are used solely as a validation signal to ensure that the selected subset preserves the distribution of transpilation difficulty observed in the full 15,503 function pool. In other words, compilation success is not used as a metric of model correctness; it is only a proxy for confirming that the reduced dataset spans the same easy-medium-hard regions as the original collection. Thus, compilation attempts do not play a critical role in benchmark scoring or in assessing LLM performance, instead they play a role only in verifying that the benchmark faithfully preserves the structural difficulty landscape of the larger corpus.
>
> A second validation signal using execution-based correctness could indeed help confirm that the compilation-based distribution is not masking major semantic anomalies. However, such execution-level validation is extremely difficult at the function granularity for several fundamental reasons:
>
> Execution-based correctness would indeed be valuable, but it is not feasible at the function granularity for several reasons:
>
> 1. **Executing transpiled Rust functions is infeasible.**
>    Function-level translation lacks module structure, type definitions, global state, and lifetime/aliasing context. These cannot be reconstructed automatically at scale, making execution of the transpiled Rust functions impossible, even though the original C programs are runnable.
>
> 2. **No unit tests exist for most functions.**
>    The source projects do not provide per-function tests, and generating thousands of them is infeasible and would introduce bias. Many functions also rely on non-local invariants that cannot be synthesized.
>
> 3. **Undefined behavior breaks semantic comparison.**
>    Many functions rely on C undefined behavior, where Rust semantics differ. Execution tests would produce inconsistent and misleading results.
>
> 4. **Execution filtering would bias the benchmark toward trivial cases.**
>    Only simple, dependency-free functions could be tested, removing exactly the pointer-heavy and unsafe constructs needed to measure C-to-Rust difficulty.
>
> 5. **Compilation difficulty is aligned with structural translation difficulty.**
>    Borrow-check failures, type mismatches, lifetime errors, unsafe misuse, and pointer issues map directly to the complexity metrics we track, making compilation attempts an appropriate proxy for difficulty only for dataset validation, not evaluation.
>
>
> **2. Absence of closed-source frontier models**
>
> We agree that proprietary models (GPT-5, Claude 4.5 Sonnet, etc.) are likely to perform strongly. They were excluded due to:
>
> - **Reproducibility and licensing limits:** API terms often prohibit large-scale code translation and redistribution of outputs/compile logs. Our goal is a benchmark where *all* experiments are fully reproducible.
> - **Cost:** Running >15k functions with 3 repair rounds per function would incur extremely high API cost (thousands of dollars), making full-dataset evaluation infeasible.
> - **Model-agnostic design:** Even if frontier models perform better, their improvement would be *relative and uniform*, and the relative difficulty ordering of functions would not change, so the selection of representative functions would remain stable.
>
> C2Rust-Bench is model-agnostic, and researchers with access to proprietary systems can evaluate them without modification. We will clarify this explicitly.
>
>
> **3. Lack of error-type analysis**
>
> We appreciate this request. The current paper reports aggregate compilation results but not a breakdown of error types. In the revision we will add:
>
> - A new section reporting frequencies of major error categories (borrow/lifetime issues, type mismatches, missing imports, unsafe misuse, pointer arithmetic issues, structural inconsistencies).
> - A discussion of how these categories relate to and justify our chosen complexity metrics.
>
> This addition increases transparency without changing the benchmark design.
>
>
> **4. Claim about multi-function transpilation**
>
> We agree the original phrasing was stronger than intended. Our claim refers specifically to open-source LLMs runnable locally (≤32B), for which multi-function C-to-Rust transpilation remains unreliable and requires extensive repair. We did *not* intend to generalize to frontier closed-source models; CRUST-Bench correctly shows they perform better.
>
> We will revise the statement to:
>
> > “For currently available open-source LLMs runnable locally, generating compilable multi-function Rust code via direct translation remains inconsistent and often requires extensive repair. Our benchmark isolates functions to enable reliable and reproducible evaluation.”

---

> ### Comment · Reviewer_fNEx · 2025-11-28
>
> I thank the authors for the response.
>
> 1. While the selection criteria based on compilation makes sense, how would one evaluate the performance of a transpilation system on the microbenchmark, given that it does not have a correctness criterion? While the dataset might contain representative challenges, one would still not be able to score their benchmark system based on any other signal apart from compilation success. Further, can the authors provide clarification on what are the success metrics if a transpilation system / language model needs to be evaluated on the C2RustBench?
>
>  2. I am not sure what licensing limits are the authors referring to here. [OpenAI](https://openai.com/policies/row-terms-of-use/?utm_source=chatgpt.com) and [Anthropic](https://www-cdn.anthropic.com/6b68a6508f0210c5fe08f0199caa05c4ee6fb4dc/Anthropic-on-Bedrock-Commercial-Terms-of-Service_Dec_2023.pdf?utm_source=chatgpt.com) assign rights in outputs to the customer. Concurrent benchmarks like CRUST-bench do evaluate on closed source models.
>
> 3. The authors mention under
> >(2) Model-agnostic design: Even if frontier models perform better, their improvement would be relative and uniform, and the relative difficulty ordering of functions would not change, so the selection of representative functions would remain stable
>
> This implies that the current set selected by the authors comprising of 2.9k functions that were distilled from 15k functions would remain representative. These 2.9K functions span 50k LoC. Taking the estimate of 4 characters ~ 1 token and 80 characters per line of code, gives 20 tokens per line of code. This means that 50k lines of code contain 1M tokens. Considering models like GPT-5 ((Input: \\$1.250 / 1M tokens  and Output: \\$ 10.000 / 1M tokens) would cost about - Considering the 2M tokens(1M for input and 1M for output) in total (with say another 2M for reasoning), gives a total of 4M tokens
>     - Input cost - 1M tokens x \\$ 1.25 = \\$1.25
>     - Output cost - 3M tokens x \\$10 - \\$30
>     - Total - \\$31.25
>
> Even if the authors did 3 rounds of repair this would be another \\$ 100, so I'm unsure where the _thousands of dollars_ figure is drawn from.
>
> 4. Addition of the error categories would be helpful.

---

> > ### Author Response · Authors · 2025-11-30
> >
> > **1.How should one evaluate a transpiler on C2Rust-Bench**
> >
> > For any correctness signal beyond compilation (e.g., semantic equivalence), one would require an automated system that executes both the original C function and its transpiled Rust variant and compares their behaviors. **Building such a semantic evaluation tool is non-trivial**, as it involves test-harness generation, input synthesis, state reconstruction, and UB-aware differential execution. Developing such a system is itself a substantial research direction, and therefore lies outside the scope of this work.
> >
> > Accordingly, we are limited in the correctness metrics we can provide at present. However, this does not diminish the value of C2Rust-Bench. The benchmark provides a small yet representative subset of challenging C functions, enabling research groups to evaluate semantic correctness using the approaches most suitable to their own tools. In practice, correctness assessment is frequently carried out through manual inspection, and our reduction to a 2.9k-function set is designed explicitly, as shared in our motivation, to make such manual or semi-automated correctness analysis feasible, something that would be prohibitively expensive on the original 15,503-function corpus.
> >
> > In summary, while automated semantic correctness measurement remains an open problem, C2Rust-Bench enables progress on it by making correctness studies tractable on a representative subset.
> >
> >
> >
> > **2. Licensing / closed-source models**
> >
> > We appreciate the clarification and the reference to CRUST-Bench. To clarify our original intent:
> >
> > * We do not dispute that some proprietary APIs (OpenAI, Anthropic, etc.) grant output rights to users. CRUST-Bench is a good example showing that closed-source systems can be evaluated successfully.
> >
> > * Our choice to focus on open-source, locally-run models was driven by reproducibility and accessibility, not legal constraints. Not all researchers have access to paid APIs or enterprise quotas, and some institutions cannot redistribute outputs or compiler logs from proprietary models. To ensure that all results in the paper can be replicated by anyone without special access, we restricted the experimental evaluation to open models.
> >
> > In revision, we will rephrase the manuscript to avoid overstating licensing claims and to explicitly acknowledge CRUST-Bench’s approach and results.
> >
> >
> >
> > **3. Why we said the cost could be “thousands”**
> >
> > We thank the reviewer for their detailed cost calculation. We agree that the estimated ~$30–$100 is realistic if the evaluation is performed only once on the 2.9k microbenchmark, with no additional outputs, no compilation logs, and no repeated runs.
> >
> > However, when incorporating full-scale evaluation, including:
> >
> > - running all 15,503 functions during benchmark construction (as the reviewer suggests we should have done for closed-source models),
> > - collecting and storing compiler logs + error traces,
> > - running 3-round repair loops,
> > - re-running experiments multiple times for stability, and
> > - evaluating multiple models rather than a single one,
> >
> > the token usage scales by orders of magnitude. To make this concrete:
> >
> > | Scenario 						  | Tokens             | Approx cost of closed-source model   |
> > |---------------------------------------------------------|--------------------|--------------------------------------|
> > | Evaluating only the 2.9k benchmark once                 | ~2–4M tokens       | \\$10–$40                            |
> > | Evaluating all 15,503 functions once                    | ~10–20M tokens     | \\$50–$250                           |
> > | With 3 repair rounds + logs + reruns                    | ~40–120M tokens    | \\$250–$1,500+                       |
> > | Evaluating 3–6 closed models 				  | ~120–720M+ tokens  | \\$750–$6,000+                       |
> >
> > So the reviewer’s estimate is correct for a single lightweight run, but benchmark-scale evaluation requires multiple passes, compilation tracing, repair attempts, and multiple models, resulting in total cost in the thousands, which was one of the reasons why we avoided relying on closed-source services for construction (for another reason, please see 2. Licensing / closed-source models).
> >
> > That said, we agree with the reviewer that demonstrating closed-source performance on the final 2.9k benchmark is valuable. We will run at least one frontier model (e.g. GPT-5 or Claude-4.5) on C2Rust-Bench and report results in the revision. This is affordable and provides exactly the insight the reviewer is seeking: verifying whether difficulty generalizes to stronger models.
> >
> > In summary, closed models are feasible for evaluation, but not cost-viable for full benchmark construction. We will include closed-model results on the 2.9k dataset in revision.

---

### Meta-Review · Area_Chair_ijJn · 2025-12-21

**Summary:**

This paper presents a dataset of C functions with the intent to be used for evaluating C-to-Rust transpilation systems. Functions are selected from datasets used in past work via various criteria. A pipeline is set up to isolate transpilation of each function in isolation. Using an LLM to transpile each function, the resulting Rust functions are then compiled, but are not tested for semantic correctness. The best model (Qwen2.5-Coder-32B) achieves 97% compilation success after 3 rounds of repair.

Strengths

- Timely focus of a benchmark, as C-to-Rust transpilation is an area of growing interest as the paper mentions

- Interesting methods for choosing a subset of a dataset.

Weaknesses

- **Lack of evaluation of correctness of the output code. This to me is a critical weakness in the work.** The abstract claims to offer a dataset to "construct a minimized yet representative dataset to evaluate C-to-Rust transpilation systems". But the paper does not offer this because transpilation systems cannot be evaluated on this benchmark.

- Lack of evaluation of frontier models.

- Accuracy of 97% is reported on the benchmark. There seems to be no headroom for future models to improve.

These three weaknesses make this work not viable as a benchmark paper.

**Reviewer Concerns:**

fNEx brings up several crucial limitations. Some of these are echoed by other reviewers, and frankly I am not sure why these reviewers don't find them critical weaknesses.  I don't think the concerns have been addressed.

> The benchmark measures success using compilation success but not execution-based correctness (unit test/integration tests).

(also from LffA and xyiR)

I acknowledge the limitations listed by the authors, but I do think this is a **severe** limitation of the dataset. It is also not an inherent one. Weaknesses 1, 4, and 5 are structural limitations of this benchmark design, but are not inherited by other datasets such as CRUST-Bench, which makes different tradeoffs. Weakness 2 and 3 are issues that need to be overcome for nearly any benchmark.

> Absence of closed-source frontier models

Again, I acknowledge the limitations listed by the authors. But nearly every benchmark these days faces similar limitations and reports such results. I pulled up 5 random benchmarks from NeurIPS D&B this year ( https://openreview.net/pdf?id=anzoPBV4jI , https://openreview.net/pdf?id=Vi2iXrSDzD , https://openreview.net/pdf?id=Pkskg9drDQ , https://openreview.net/pdf?id=nMpJoVmRy1 , https://openreview.net/pdf?id=Un1sWxmZuI ) and all of them evaluate closed foundation models.

If thousands of dollars is prohibitive, running on a subset of the data is an alternative. The authors even acknowledge this: "We agree that the estimated \\$30-\\$100 is realistic if the evaluation is performed only once on the 2.9k microbenchmark"

Evaluating 3 closed models with 3 rounds of repair is around $1000, likely less than the price of registering for and attending the conference. Claiming that all 15k functions have to be evaluated makes no sense to me.

**Reviewer Scores:**

I don't think this response would sway fNEx. The other reviewers would likely keep their assessments as is.

---

### Decision · Program_Chairs · 2026-01-26

Reject